# Ontology-driven weak supervision for clinical entity classification in electronic health records

Jason A. Fries 1✉, Ethan Steinberg[1,2], Saelig Khattar[2], Scott L. Fleming [1], Jose Posada [1], Alison Callahan [1] & Nigam H. Shah [1]

In the electronic health record, using clinical notes to identify entities such as disorders and their temporality (e.g. the order of an event relative to a time index) can inform many important analyses. However, creating training data for clinical entity tasks is time consuming and sharing labeled data is challenging due to privacy concerns. The information needs of the COVID-19 pandemic highlight the need for agile methods of training machine learning models for clinical notes. We present Trove, a framework for weakly supervised entity classification using medical ontologies and expert-generated rules. Our approach, unlike hand-labeled notes, is easy to share and modify, while offering performance comparable to learning from manually labeled training data. In this work, we validate our framework on six benchmark tasks and demonstrate Trove's ability to analyze the records of patients visiting the emergency department at Stanford Health Care for COVID-19 presenting symptoms and risk factors.

---

[1] Center for Biomedical Informatics Research, Stanford University, Stanford, CA, USA. [2] Department of Computer Science, Stanford University, Stanford, CA, USA. ✉email: jason-fries@stanford.edu

Analyzing text to identify concepts such as disease names and their associated attributes like negation are foundational tasks in medical natural language processing (NLP). Traditionally, training classifiers for named entity recognition (NER) and cue-based entity classification have relied on hand-labeled training data. However annotating medical corpora requires considerable domain expertise and money, creating barriers to using machine learning in critical applications[1,2]. Moreover, hand-labeled datasets are static artifacts that are expensive to change. The recent COVID-19 pandemic highlights the need for machine-learning tools that enable faster, more flexible analysis of clinical and scientific documents in response to rapidly unfolding events[3].

To address the scarcity of hand-labeled training data, machine-learning practitioners increasingly turn to lower cost, less accurate label sources to rapidly build classifiers. Instead of requiring hand-labeled training data, weakly supervised learning relies on task-specific rules and other imperfect labeling strategies to programmatically generate training data. This approach combines the benefits of rule-based systems, which are easily shared, inspected, and modified, with machine learning, which typically improves performance and generalization properties. Weakly supervised methods have demonstrated success across a range of NLP and other settings[4–8].

Knowledge bases and ontologies provide a compelling foundation for building weakly supervised entity classifiers. Ontologies codify a vast amount of medical knowledge via taxonomies and example instances for millions of medical concepts. However, repurposing ontologies for weak supervision creates challenges when combining label information from multiple sources without access to ground-truth labels. The hundreds of terminologies found in the Unified Medical Language System (UMLS) Metathesaurus[9] and other sources[10] typify the highly redundant, conflicting, and imperfect entity definitions found across medical ontologies. Naively combining such conflicting label assignments can cause substantial performance drops in weakly supervised classification[11]; therefore, a key challenge is correcting for labeling errors made by individual ontologies when combining label information.

Rule-based systems for NER and cue detection[12,13] are common in clinical text processing, where labeled corpora are difficult to share due to privacy concerns. Generating imperfect training labels from indirect sources (e.g., patient notes) is often used in analyzing medical images[14–16]. Recent work has explored learning the accuracies of sources to correct for label noise when using rule-based systems to generate training data for text

classification[4,17]. Weakly supervised clinical applications have explored document and relation classification using task-specific rules[18,19] or leveraging dependency parsing and compositional grammars to automate relation classification for standardizing clinical concepts[20]. However, these largely focus on relation and document classification via task-specific labeling rules or sourcing supervision from a single ontology and do not explore NER or automating labeling via multiple ontologies.

Prior research on weakly supervised NER has required complex preprocessing to identify possible entity spans[21], generated labels from a single source rather than combining multiple sources[22], or relied on ad hoc rule engineering[23]. High-impact application areas, such as clinical NER using weak supervision, are largely unstudied. Recent weak supervision frameworks such as Snorkel[11] are domain- and task-agnostic, introducing barriers to quickly developing and deploying labeling heuristics in complex domains such as medicine. Key questions remain about the extent to which we can automate weak supervision using existing medical ontologies and how much additional task-specific rule engineering is required for state-of-the-art performance. It is also unclear whether, and by how much, pretrained language models such as BioBERT[24] improve the ability to generalize from weakly labeled data and reduce the need for task-specific labeling rules.

We present a Trove, a framework for training weakly supervised medical entity classifiers using off-the-shelf ontologies as a source of reusable, easily automated labeling heuristics. Doing so transforms the work of using weak supervision from that of coding task-specific labeling rules to defining a target entity type and selecting ontologies with sufficient coverage for a target dataset, which is a common interface for popular biomedical annotation tools such as NCBO BioPortal and MetaMap[10,25]. We examine whether ontology-based weak supervision, coupled with recent pretrained language models such as BioBERT, reduces the engineering cost of creating entity classifiers while matching the performance of prior, more expensive, weakly supervised approaches. We further investigate how ontology-based labeling functions can be extended when we need to incorporate additional, task-specific rules. The overall pipeline is shown in Fig. 1.

In this work, we demonstrate the utility of Trove through six benchmark tasks for clinical and scientific text, reporting state-of-the-art weakly supervised performance (i.e., using no hand-labeled training data) on NER datasets for chemical/disease and drug tagging. We further present weakly supervised baselines for two tasks in clinical text: disorder tagging and event temporality classification. Using ablation analyses, we characterize the performance trade-offs of training models with labels generated from

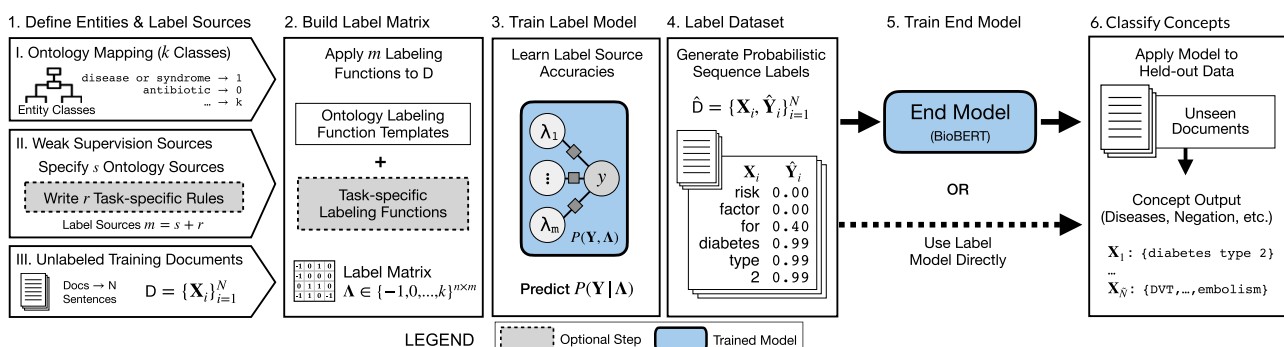

**Fig. 1 Trove pipeline for ontology-driven weak supervision for medical entity classification.** Users specify (I) a mapping of an ontology's class categories to entity classes, (II) a set of label sources (e.g., ontologies, task-specific rules) for weak supervision, and (III) a collection of unlabeled document sentences with which to build a training set. Ontologies instantiate labeling function templates that are applied to sentences to generate a label matrix. This matrix is used to train the label model that learns source accuracies and corrects for label noise to predict a consensus probability per word. Consensus labels are transformed into the probabilistic sequence label dataset that is used as training data for an end model (e.g., BioBERT). Alternatively, the label model can also be used as the final classifier.

**Table 1 F1 scores for ontology and task-specific rule-based weak supervision.**

| Task | Ontologies (guidelines + UMLS + other) | | | | +Task-specific rules | | | | Hand-labeled | |
|---|---|---|---|---|---|---|---|---|---|---|
| | LFs | MV | LM | WS | LFs | MV | LM | WS | FS | SOTA |
| Chemical | 22 | 79.8 | 88.0 ± 0.1† | **88.5 ± 0.2*** | +9 | 81.1 | 89.2 ± 0.2† | **91.1 ± 0.1*** | 92.4 ± 0.2 | 93.5[24] |
| Disease | 16 | 74.7 | **78.9 ± 0.1†** | 78.3 ± 0.2* | +6 | 76.4 | 79.8 ± 0.3† | **79.9 ± 0.2** | 84.5 ± 0.2 | 87.2[24] |
| Disorder | 25 | 67.8 | 68.3 ± 0.3† | **69.1 ± 0.2*** | +11 | 71.2 | 75.0 ± 0.2† | **76.3 ± 0.1*** | 79.6 ± 0.3 | 80.1[65] |
| Drug | 16 | 75.3 | 78.6 ± 0.1† | **79.2 ± 0.2*** | +11 | 82.2 | 85.8 ± 0.4† | **88.3 ± 0.3*** | 93.2 ± 0.3 | 91.4[66] |
| Negation | – | – | – | – | 17 | 92.5 | **93.0 ± 0.0†** | 92.7 ± 0.6* | 96.1 ± 0.2 | ~ |
| DocTimeRel | – | – | – | – | 27 | 67.8 | 69.2 ± 0.0† | **72.9 ± 0.5*** | 86.2 ± 0.1 | 83.4[67] |

Models are majority vote (MV); label model (LM); weakly supervised BioBERT (WS); our fully supervised BioBERT (FS); and published state-of-the-art (SOTA). LFs denote labeling function counts or total added task-specific rules. Bold indicates the best score for each approach and task. Scores are the mean and ±1 SD of $n = 10$ random weight initializations. A two-sided Wilcoxon signed-rank test was used to compute statistical significance. *Denotes $P < 0.05$ for the difference between weakly supervised BioBERT (WS) and the label model (LM). For (chemical, disease, disorder, drug) exact $P$ values for ontologies were (0.0039, 0.0020, 0.0020, 0.0020) and for task-specific rules (0.0020, 0.3223, 0.0020, 0.0020). For Negation $P = 0.0273$ and for DocTimeRel $P = 0.0020$. †Denotes $P < 0.05$ for the difference between the label model (LM) and majority vote (MV). Here, all task $P$ values were 0.0020. ~Mowery et al.[68] only reported accuracy for the negation task.

easily automated ontology-based weak supervision vs. more expensive, task-specific rules. Finally, we present a case study deploying Trove for COVID-19 symptom tagging and risk factor monitoring using a daily data feed of Stanford Health Care emergency department notes.

Weakly supervised learning is an umbrella term referring to methods for training classifiers using imperfect, indirect, or limited labeled data and includes techniques such as distant supervision[26,27], co-training[28], and others[29]. Prior approaches for weakly supervised NER such as cotraining use a small set of labeled seed examples[30] that are iteratively expanded through bootstrapping or self-training[31]. Semi-supervised methods also use some amount of labeled training data and incorporate unlabeled data by imposing constraints on properties such as expected label distributions[32]. Distant supervision requires no labeled training data, but typically focuses on a single source for labels such as AutoNER[22], which used phrase mining and a tailored dictionary of canonical entity names to construct a more precise labeler, rather than unifying labels assigned using heterogeneous sources of unknown quality. Crowdsourcing methods combine labels from multiple human annotators with unknown accuracy[33]. However, compared to human labelers, programmatic label assignment has different correlation and scaling properties that create technical challenges when combining sources. Data programming[11,17,34] formalizes theory for combining multiple label sources with different coverage and unknown accuracy, as well as correlation structure to correct for labeling errors.

In the setting of weakly supervised NER and sequence labeling, SwellShark[21] uses a variant of data programming to train a generative model using labels from multiple dictionary and rule-based sources. However, this approach required task-specific preprocessing to identify candidate entities a priori to achieve competitive performance. Safranchik et al.[23] presented WISER, a linked hidden Markov model where weak supervision was defined separately over tags and tag transitions using linking rules derived from language models, ngram statistics, mined phrases, and custom heuristics to train a BiLSTM-CRF. SwellShark and WISER both focused on hand-coded, task-specific labeling function design.

Trove advances weakly supervised medical entity classification by (1) eliminating the requirement for identifying probable entity spans a priori by combining word-level weak supervision with contextualized word embeddings, (2) developing general-purpose, more easily automated ontology-based labeling functions that reduce the need for engineering hand-coded rules, (3) quantifying the relative contributions of sources of label assignment—such as pre-existing ontologies from the UMLS (low cost) and task-specific rule engineering (high cost)—to the achieved

performance for a task, and (4) evaluating Trove in a deployed medical setting, tagging symptoms and risk factors of COVID-19.

## Results

**Experiment overview.** After quantifying the performance of ontology-driven weak supervision in all our tasks, we performed four experiments. First, we examined performance differences by label source ablations, which compared ontology-based labeling functions against those incorporating task-specific rules. Second, we compared Trove to existing weakly supervised tagging methods. Third, we examined learning source accuracies for UMLS terminologies. Finally, we report on a case study that used Trove to monitor emergency department notes for symptoms and risk factors associated with patients tested for COVID-19.

We evaluated four methods of combining labeling functions to train entity classifiers. (1) Majority vote (MV) is the majority class for each word predicted by all labeling functions. In cases of abstain or ties, predictions default to the majority class. (2) Label model (LM) is the default data programming model. Abstain and ties default to the majority class. (3) Weakly supervised (WS) is BioBERT trained on the probabilistic dataset generated by the label model. (4) Fully supervised (FS) is BioBERT trained on the original expert-labeled training set, tuned to match current published state-of-the-art performance, and using the validation set for early stopping.

For reference, we also included published F1 metrics for state-of-the-art (SOTA) supervised performance for each task, as determined to the best of our knowledge. Note that some published SOTA benchmarks (e.g., BC5CDR in Lee et al.[24]) use both the hand-labeled train and validation sets for training, so they are not directly comparable to our experimental setup.

**Performance of Trove in medical entity classification tasks.** Table 1 reports F1 performance for weak supervision using ontology-based labeling functions and those incorporating additional, task-specific rules. For NER tasks, adding task-specific rules performed within 1.3–4.9 F1 points (4.1%) of models trained on hand-labeled data and for span tasks within 3.4–13.3 F1 points. The total number of task-specific labeling functions used ranged from 9 to 27. For ontology-based supervision, the label model improved performance over MV by 4.1 F1 points on average, and BioBERT provided an additional average increase of 0.3 F1 points.

**Labeling source ablations.** For NER tasks, we examined five ablations, ordered by increasing the cost of labeling effort. (1) Guidelines, a dictionary of all positive and negative examples explicitly provided in annotation guidelines, including

dictionaries for punctuation, numbers, and English stopwords. (2) +UMLS, all terminologies available in the UMLS. (3) +Other, additional ontologies or existing dictionaries not included in the UMLS. (4) +Rules, task-specific rules, including regular expressions, small dictionaries, and other heuristics. (5) Hand-labeled, supervised learning using the expert-labeled training split.

Tiers 1–4 are additive and include all prior levels. We initialized labeling function templates as follows:

For ontology-based labeling functions, we used the UMLS Semantic Network and the corresponding Semantic Groups as our entity categories and defined a mapping of semantic types (STYs) to target class labels $y \in \{-1, 0, 1\}$. Non-UMLS ontologies that did not provide semantic-type assignments (e.g., ChEBI) were mapped to a single class label. All UMLS terminologies $v$ were ranked by term coverage on the unlabeled training set, defined as each term's document frequency summed by terminology, and the top $s$ terminologies were used to initialize templates, where $s$ was tuned with a validation set. The remaining $(v_{s+1}, \ldots, v_{92})$ UMLS terminologies were merged into a single labeling function to ensure that all terms in the UMLS were included. UMLS synsets were constructed using concept unique identifiers (CUIs) and templates were initialized with the union of all terminologies and fixed across all NER tasks.

For task-specific labeling functions, we evaluated our ability to supplement ontology-based supervision with hand-coded labeling functions and estimated the relative performance contribution of adding these task-specific rules. All training set documents were preprocessed to tag entities using the ontology-based labeling functions outlined above and indexed to support search queries for efficient data exploration. The design of task-specific labeling functions is a mix of data exploration, i.e., looking at entities identified by ontology labeling functions to identify errors, and similarity search to identify common, out-of-ontology concept patterns. Only the training set was examined during this process and the test set was held out during all labeling function development and model tuning.

For NER, we used two rule types to label concepts: (1) pattern matching via regular expressions and small dictionaries of related terms (e.g., illegal drugs), and (2) bigram word co-occurrence graphs from ontologies to support fuzzy span matching. Pattern matching comprised the majority of our task-specific labeling functions. While task-specific labeling functions codify generalized patterns not captured by ontologies, we also note that a number of our task-specific labeling functions were necessary due to the idiosyncratic nature of ground-truth labels in benchmark tasks. For example, in the i2b2/n2c2 drug tagging task, annotation guidelines included more complex, conditional entity definitions, such as not labeling negated or historical drug mentions. We incorporated these guidelines using the Negation and DocTimeRel labeling functions described below. See Supplementary Fig. 1 and Supplementary Note for a more detailed example of designing task-specific labeling functions.

For span tasks, which classify Negation and DocTimeRel for preidentified entities, we do not use ontology-based labeling functions directly for supervision. Instead, ontology-tagged entities were used to guide the development of labeling functions that search left- and right-context windows around a target entity for cue phrases. Designing search patterns for left- and right-context windows is the same strategy used by NegEx/ConText[12,35] to assign negation and temporal status. For Negation, we built on NegEx by adding additional patterns found via exploration of the training documents. For DocTimeRel, we used a heuristic based on the nearest explicit datetime mention (in the token distance) to an event mention[36]. Additional contextual pattern matching rules were added to detect other cues of event temporality, e.g., using section headers such as past medical history to identify events occurring before the note creation time.

Figure 2 reports F1 scores across all ablation tiers. In all settings, the weakly supervised BioBERT models outperformed MV. Gains of 8.0–34.7 F1 points are seen in the guideline-only tier and 1.3–8.2 points in other tiers. Incorporating source

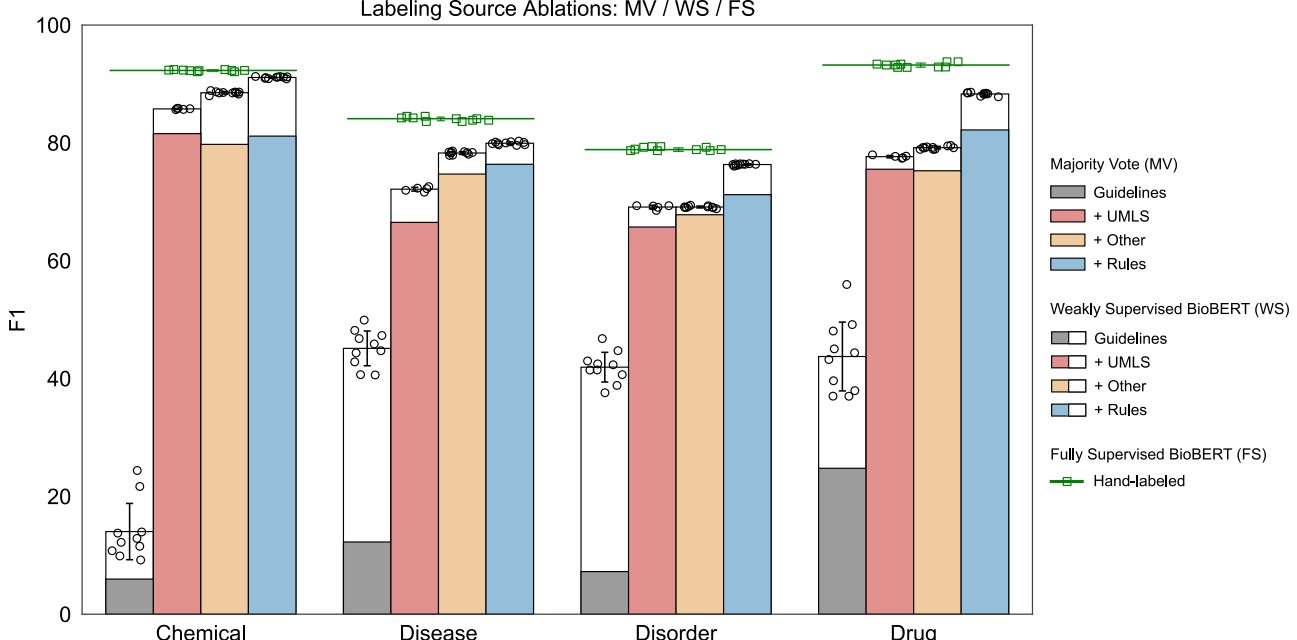

**Fig. 2 Ablation study of F1 performance by labeling source.** Majority vote (MV) vs. weakly supervised BioBERT (WS) vs. fully supervised (FS) for all labeling source ablations showing the absolute F1 score for all labeling tiers. The colored region of each bar indicates MV performance and the white regions denote performance improvements of WS over MV. The mean performance of FS is indicated by the green lines and square points. WS and FS consist of $n = 10$ experiment replicates using different random initialization seeds, presented as the mean with error bars ± SD. MV is deterministic and does not include replicates.

accuracies into BioBERT training provided significant benefits when combining high-precision sources with low-precision/high-recall sources. In the case of chemical tagging with MV, the UMLS tier (red) outperformed UMLS+Other (orange) by 1.8 F1 points (81.6 vs. 79.8). This was due to adding the ChEBI ontology that increased recall but only had 65% word-level precision. Majority vote cannot learn or utilize this information, so naively adding ChEBI labels hurts performance. However, the label model learned ChEBI's accuracy to take advantage of the noisier, but higher-coverage signal; thus, the WS UMLS + Other (orange + white) outperformed UMLS ((red + white)) by 2.5 F1 points (88.0 vs. 85.5). See Supplementary Tables 1–4 for complete performance metrics across all ablation tiers.

**Comparing Trove with existing weakly supervised methods.** We compared Trove to three existing weakly supervised methods for NER and sequence labeling: SwellShark[21], AutoNER[22], and WISER[23]. We compared performance on BC5CDR (the combination of disease and chemical tasks) against all methods and on the i2b2/n2c2 drug task for Swell-Shark. All performance numbers are for models trained on the original training set split, with the exception of SwellShark that is trained on an additional 25,000 weakly labeled documents. All weakly supervised methods use the labeling functions, preprocessing, and dictionary curation methods as described in the original manuscripts. Table 2 compares Trove with these existing weakly supervised methods. Our ontology-based approach outperformed AutoNER by 1.7 F1 points. For models incorporating task-specific rules, we outperformed the best weakly supervised model SwellShark by 1.9 F1 points. Swell-Shark reported F1 scores on the i2b2/n2c2 drug task of 78.3 for dictionaries and 83.4 for task-specific rules. Our best models achieved 79.2 and 88.4 F1, respectively.

**UMLS terminologies as plug-and-play weak supervision.** Biomedical annotators such as NCBO BioPortal require selecting a set of target ontologies/terminologies to use for labeling. Since Trove is capable of automatically combining noisy terminologies, given a shared semantic-type definition, we tested the ability to avoid selecting specific UMLS terminologies for use as supervision sources. This is challenging because estimating accuracies with the label model requires observing agreement and disagreement among multiple label sources; however, it is non-obvious how to partition the UMLS, which contains many terminologies, into labeling functions. The naive extremes are to either create a single labeling function from the union of all terminologies or include all terminologies as individual labeling functions.

To explore how partitioning choices impact label model performance, we held all non-UMLS labeling functions fixed across all ablation tiers and computed performance across $s = (1, \ldots, 92)$ partitions of the UMLS by terminology. All scores were normalized to the best global majority vote score per tier, selected using the best $s$ choice evaluated on the validation set, to assess the impact of correcting for label noise.

Figure 3 shows the impact of partitioning the UMLS into $s$ different labeling functions. Modeling source accuracy consistently outperformed MV across all tiers, in some cases by 2–8 F1 points. The best performing partition size $s$ ranged from 1 to 10 by task. The naive baseline approaches—collapsing the UMLS into a single labeling function or treating all terminologies as individual labeling functions—generally did not perform best overall.

**Case study in rapidly building clinical classifiers.** We deployed Trove to monitor emergency departments for patients undergoing COVID-19 testing, analyzing clinical notes for presenting symptoms/disorders and risk factors[37]. This required identifying disorders and defining a novel classification task for exposure to a confirmed COVID-19-positive individual, a risk factor informing patient contact tracing. The dataset consisted of daily dumps of emergency department notes from Stanford Health Care (SHC), beginning in March 2020. Our study was approved by the Stanford University Administrative Panel on Human Subjects Research, protocol #24883, and included a waiver of consent. All included patients from SHC signed a privacy notice, which informs them that their records may be used for research purposes given approval by the IRB, with study procedures in place to protect patient confidentiality.

We manually annotated a gold test set of 20 notes for all mentions of disorders and 776 notes for mentions of positive COVID exposure. Two clinical experts generated gold annotations that were adjudicated for disagreements by authors AC and JAF. As a baseline for disorder tagging, we used the fully supervised ShARe/CLEF disorder tagger. This reflects a readily available, but out-of-distribution training set (MIMIC-II[38] vs. SHC). We used the same disorder labeling function set as our prior experiments, adding one additional dictionary of COVID terms[39]. BioBERT was trained using 2482 weakly labeled documents. Custom labeling functions were written for the exposure task and models were trained on 14 k sentences.

Table 3 contains our COVID case study results. The label model provided up to 5.2 F1 points improvement over majority vote and performed best overall for disorder tagging. Our best weakly supervised model outperformed the disorder tagger trained on hand-labeled MIMIC-II data by 2.3 F1 points. For exposure classification, the label model provided no benefit, but the weakly supervised end model provided a 6.9% improvement (+5.2 F1 points) over the rules alone.

---

**Table 2 Comparison of Trove against existing weakly supervised NER methods.**

| Supervision method | Label source | Number of train docs | End model | P | R | F1 |
|---|---|---|---|---|---|---|
| Fully supervised | Hand-labeled | 500 | BioBERT | 87.6 | 89.3 | 88.7 |
| Fully supervised | Hand-labeled | 500 | BiLSTM-CRF | 87.2 | 87.9 | 87.5 |
| SwellShark | Dictionaries | 25,500 | BiLSTM-CRF | <u>84.6</u> | 74.1 | 79.0 |
| AutoNER | Dictionaries | 500 | BiLSTM-CRF | <u>83.2</u> | 81.1 | 82.1 |
| Ours (Trove + Snorkel) | Dictionaries | 500 | BioBERT | 81.6 | <u>86.1</u> | <u>83.7</u> |
| SwellShark | Custom rules | 25,500 | BiLSTM-CRF | **86.1** | 82.4 | 84.2 |
| WISER | Custom rules | 500 | BiLSTM-CRF | 82.7 | 83.3 | 83.0 |
| Ours (Trove + Snorkel) | Custom rules | 500 | BioBERT | 85.5 | **86.8** | **86.1** |

Precision (P), recall (R), and F1 scores for the BC5CDR task. Underlined numbers indicate the best weakly supervised score using only dictionaries/ontologies, and bold indicates the best score using custom rules. For this task, ontology-based supervision alone outperformed existing weakly supervised methods except for SwellShark which required custom rules and candidate generation. Incorporating task-specific rules into Trove further improved performance.

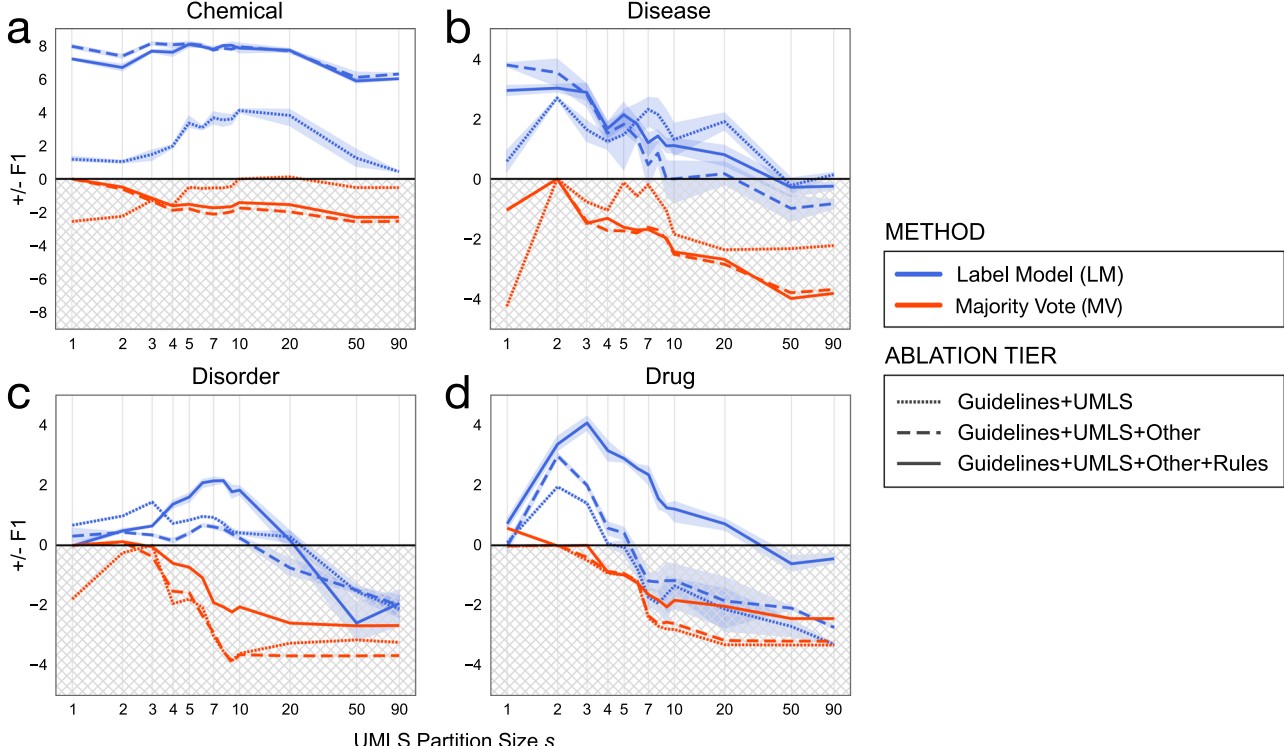

**Fig. 3 The relationship between the number of UMLS partitions and the learned accuracies of label sources. a** BC5CDR chemical entities. **b** BC5CDR disease entities. **c** ShARe/CLEF 2014 disorder entities. **d** i2b2/n2c2 2009 drug entities. The UMLS is partitioned into *s* terminologies (*x* axis, log-scale) ordered by term coverage on the unlabeled training set. Red (MV) and blue (LM) lines are the mean difference in F1 performance (*y* axis) of *n* = 5 random weight initializations. Error bars are represented using the solid colored line to denote the mean value of data points and the shaded regions corresponding to ± SD. The gray region indicates performance worse than the best possible MV, discovered via the validation set. Across virtually all partitioning choices, modeling source accuracies outperformed MV, with *k* = 1–10 performing best overall.

**Table 3 COVID-19 presenting symptoms/disorders and risk factors evaluated on Stanford Health Care emergency department notes.**

| Supervision | Task | MV | | | LM | | | WS | | | FS | | |
|---|---|---|---|---|---|---|---|---|---|---|---|---|---|
| | | P | R | F1 | P | R | F1 | P | R | F1 | P | R | F1 |
| Hand-labeled | Disorder | – | | | – | | | – | | | 68.0 | 74.5 | 71.1 |
| Ontologies | Disorder | 64.4 | 66.4 | 65.3 | 69.3 | 71.7 | 70.5 | 67.1 | 72.3 | 69.6 | – | | |
| +Task-specific | Disorder | 69.1 | 70.4 | 69.8 | **73.0** | 73.9 | **73.4** | 70.5 | **74.8** | 72.6 | – | | |
| Task-specific | Exposure | 82.6 | 69.1 | 75.2 | 82.6 | 69.1 | 75.2 | <u>87.2</u> | <u>74.5</u> | <u>80.4</u> | – | | |

Bold and underlined scores indicate the best score in symptom/disorder tagging and COVID exposure classification respectively. Ontology-based weak supervision performed almost as well as the out-of-distribution, hand-labeled MIMIC-II data used for FS. Adding task-specific rules, even though they were developed without seeing Stanford data, outperformed the hand-labeled FS model by 2.3 F1 points.

## Discussion

Our experiments demonstrate the effectiveness of using weakly supervised methods to train entity classifiers using off-the-shelf ontologies and without requiring hand-labeled training data. Medical ontologies are freely available sources of weak supervision for NLP applications[40], and in several NER tasks, our ontology-only weakly supervised models matched or outperformed more complex weak supervision methods in the literature. Our work also highlights how domain-aware language models, such as BioBERT, can be combined with weak supervision to build low-cost and highly performant medical NLP classifiers.

Rule-based approaches are common tools in scientific literature analysis and clinical text processing[41]. Our results suggest that engineering task-specific rules in addition to labels provided by ontologies provides strong performance for several NER tasks—in

some cases approaching the performance of systems built using hand-labeled data. We further demonstrated how leveraging the structure inherent in knowledge bases such as the UMLS to estimate source accuracies and correct for label noise provides substantial performance benefits. We find that the classification performance of the label model alone is strong, with BioBERT providing modest gains of 1.0 F1 points on average. Since the label model is orders of magnitude more computationally efficient to train than BERT-based models, in many settings (e.g., limited access to high-end GPU hardware), the label model alone may suffice.

Our tasks reflect a wide range of difficulties. Clinical tasks required more task-specific rules to address the increased complexity of entity definitions and other nongrammatical, sub-language phenomena[42]. Here custom rules improved clinical tasks an average of 8.1 F1 points vs. 2.1 points for scientific

**Table 4 Dataset summary statistics.**

| Task | Domain | Name | Type | k | Documents | Entities |
|---|---|---|---|---|---|---|
| Disease | Literature | BC5CDR[69] | NER | 2 | 500/500/500 | 4182/4244/4424 |
| Chemical | Literature | BC5CDR[69] | NER | 2 | 500/500/500 | 5203/5347/5385 |
| Disorder | Clinical | ShARe/CLEF 2014[68] | NER | 2 | 166/133/133 | 5619/4449/7367 |
| Drug | Clinical | i2b2/n2c2 2009[70] | NER | 2 | 100/75/75 | 3157/2504/2819 |
| Negation | Clinical | ShARe/CLEF 2014[68] | Span | 2 | 166/133/133 | 5619/4449/7367 |
| DocTimeRel | Clinical | THYME 2016[71] | Span | 4 | 293/147/151 | 38937/20974/18990 |

There are (k) classes per task. The (Documents) and (Entities) columns indicate counts for train/validation/test splits.

literature. Moreover, adding non-UMLS ontologies to PubMed tasks consistently improved overall performance while providing little-to-no benefit for our clinical tasks. Annotation guidelines for our clinical tasks also increased complexity. The i2b2/n2c2 drug task combines several underlying classification problems (e.g., filtering out negated medications, patient allergies, and historical medications) into a single tagging formulation. This extends beyond entity typing and requires a more complex cue-driven rule design.

Manually labeling training data is time-consuming and expensive, creating barriers to using machine learning for new medical classification tasks. Sometimes, there is a critical need to rapidly analyze both scientific literature and unstructured electronic health record data—as in the case of the COVID-19 pandemic when we need to understand the full repertoire of symptoms, outcomes, and risk factors at short notice[37,43,44]. However, sharing patient notes and constructing labeled training sets presents logistical challenges, both in terms of patient privacy and in developing infrastructure to aggregate patient records[45]. In contrast, labeling functions can be easily shared, edited, and applied to data across sites in a privacy-preserving manner to rapidly construct classifiers for symptom tagging and risk factor monitoring.

This work has several limitations. Our task-specific labeling functions were not exhaustive and only reflect low-cost rules easily generated by domain experts. Additional rule development could lead to improved performance. In addition, we did not explore data augmentation or multitask learning in the BioBERT model, which may further mitigate the need to engineer task-specific rules. There is considerable prior work developing machine-learning models for tagging disease, drug, and chemical entities that could be incorporated as labeling functions. However, our goal was to explore performance trade-offs in settings where existing machine-learning models are not available. Our framework leverages the wide range of medical ontologies available for English language settings, which provides considerable advantages for weakly supervised methods. Additional work is needed to characterize the extent to which the framework can benefit tasks in non-English settings. Combining labels from multiple ontology sources violates an independence assumption of data programming as used in this work, because for any pair of source ontologies, we may have correlated noise. This restriction applies to all label sources but is more prevalent in cases with extremely similar label sources, as can occur with ontologies. In our experiments, for a small number of sources, the impact was minor; however, performance tended to decrease after including more than 20 ontologies. Additional research into unsupervised methods for structure learning[46,47], i.e., learning dependencies among sources from unlabeled data, could further improve performance or mitigate the need to limit the number of included ontologies.

Identifying named entities and attributes, such as negation, are critical tasks in medical natural language processing. Manually labeling training data for these tasks is time consuming and expensive, creating a barrier to building classifiers for new tasks. The Trove framework provides ontology-driven weak supervision for medical entity classification and achieves state-of-the-art weakly supervised performance in the NER tasks of recognizing chemicals, diseases, and drugs. We further establish new weakly supervised baselines for disorder tagging and classifying the temporal order of an event entity relative to its document time-stamp. The weakly supervised NER classifiers perform within 1.3–4.9 F1 points of classifiers trained with hand-labeled data. Modeling the accuracies of individual ontologies and rules to correct for label noise improved performance in all of our entity classification tasks. Combining pretrained language models such as BioBERT with weak supervision results in an additional improvement in most tasks.

The Trove framework demonstrates how classifiers for a wide range of medical NLP tasks can be quickly constructed by leveraging medical ontologies and weak supervision without requiring manually labeled training data. Weakly supervised learning provides a mechanism for combining the generalization capabilities of state-of-the-art machine learning with the flexibility and inspectability of rule-based approaches.

## Methods

**Datasets and tasks**. We analyze two categories of medical tasks using six datasets: (1) NER and (2) span classification where entities are identified a priori and classified for cue-driven attributes such as negation or document relative time, i.e., the order of an event entity relative to the parent document's timestamp. Both categories of tasks are formalized as token classification problems, either tagging all words in a sequence (NER) or just the head words for an entity set (span classification). Table 4 contains summary statistics for all six datasets. All documents were preprocessed using a spaCy[48] pipeline optimized for biomedical tokenization and sentence boundary detection[19].

Our COVID-19 case study used a daily feed of emergency department notes from Stanford Health Care (SHC), beginning in March 2020. Our study was approved by the Stanford University Administrative Panel on Human Subjects Research, protocol #24883 and included a waiver of consent. All included patients from SHC signed a privacy notice which informs them that their records may be used for research purposes given approval by the IRB, with study procedures in place to protect patient confidentiality.

We used 99 label sources covering a broad range of medical ontologies. We used the 2018AA release of the UMLS Metathesaurus, removing non-English and zoonotic source terminologies, as well as sources containing fewer than 500 terms, resulting in 92 sources. Additional sources included the 2019 SPECIALIST abbreviations[49], Disease Ontology[50], Chemical Entities of Biological Interest (ChEBI)[51], Comparative Toxicogenomics Database (CTD)[52], the seed vocabulary used in AutoNER[22], ADAM abbreviations database[53], and word sense abbreviation dictionaries used by the clinical abbreviation system CARD[54].

We applied minimal preprocessing to all source ontologies, filtering out English stopwords and numbers, applying a letter case normalization heuristic to preserve abbreviations, and removing all single-character terms. We did not incorporate UMLS term-type information, such as filtering out terms explicitly denoted as suppressible within a terminology since this information is not typically available in non-UMLS ontologies. Our overall goal was to impose as few assumptions as possible when importing terminologies, evaluating their ability to function as plug-and-play sources for weak supervision.

**Formulation of the labeling problem**. We assume a sequence-labeling problem formulation, where we are given a dataset $D = \{\mathbf{X}_i\}_{i=1}^{N}$ of $N$ sequences

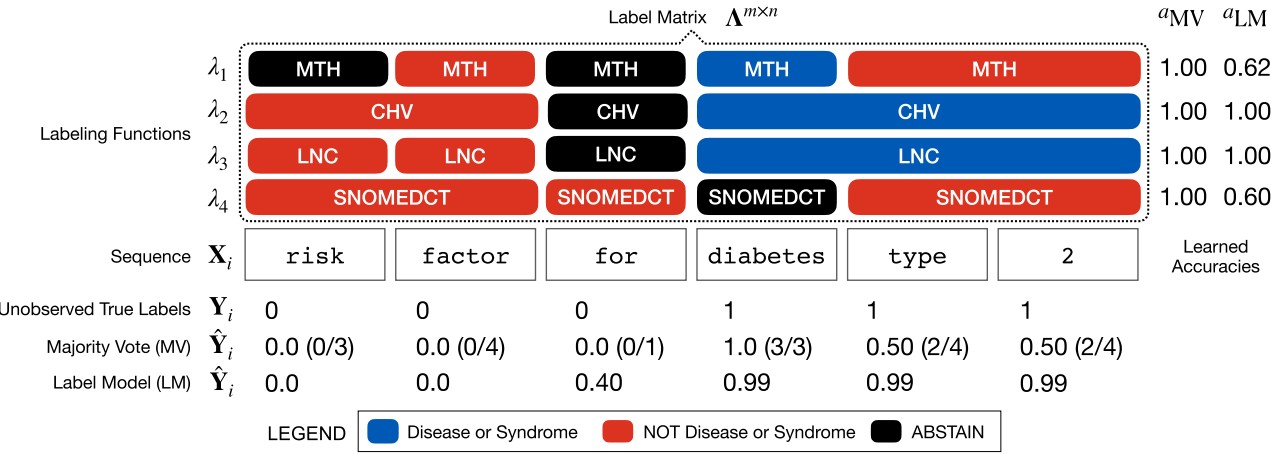

**Fig. 4 An example of combining ontology-based labeling functions.** Here four ontology labeling functions (MTH, CHV, LNC, SNOMEDCT) are used to label a sequence of words $\mathbf{X}_i$ containing the entity diabetes type 2. Majority vote estimates $\mathbf{Y}_i$ as a word-level sum of positive class labels, weighing each equally ($a_{MV}$). The label model learns a latent class-conditional accuracy ($a_{LM}$) for each ontology, which is used to reweight labels to generate a more accurate consensus prediction of $\mathbf{Y}_i$.

$\mathbf{X}_i = (x_{i,1}, \ldots, x_{i,t})$ consisting of words $x$ from a fixed vocabulary. Each sequence is mapped to a corresponding sequence of latent class variables $\mathbf{Y}_i = (y_{i,1}, \ldots, y_{i,t})$, where $y \in \{0, \ldots, k\}$ for $k$-tag classes. Since $\mathbf{Y}$ is not observable, our primary technical challenge is estimating $\mathbf{Y}$ from multiple, potentially conflicting label sources of unknown quality to construct a probabilistically labeled dataset $\hat{D} = \{\mathbf{X}_i, \widehat{\mathbf{Y}}_i\}_{i=1}^N$. This dataset can then be used for training classification models such as deep neural networks. Such a labeling regimen is typically low-cost, but less accurate than the hand-curated labels used in traditional supervised learning; hence, this paradigm is referred to as weakly supervised learning.

**Unifying and denoising sources with a label model.** When using biomedical annotators such as MetaMap or NCBO BioPortal, users specify a target set of entity classes and a set of terminology sources with which to generate labeled concepts. Consider the example outlined in Fig. 4, where we want to train an entity tagger for disease names using labels generated from four terminologies. Here, we are interested in generating a consensus set of entities using each terminology's labeled output. A straightforward unification method is majority vote

$$\hat{y} = \arg\max_{y \in \{1,\ldots,k\}} \sum_{i=1}^m \mathbb{1}_k(\lambda_i(x) = y) \tag{1}$$

where our $m$ terminologies are represented as individual labeling functions $\lambda_i$. Labeling functions encode an underlying heuristic such as matching strings against a dictionary and given an input instance (e.g., a document or entity span) assign a label in the domain $\{-1, 0, \ldots, k\}$ where $-1$ denotes abstain, i.e., not assigning any class label. Majority vote simply takes the mode of all labeling function outputs for each word, emitting the majority class in the case of ties or abstains.

Majority vote weights source equally when combining labels, an assumption that does not hold in practice, which introduces noise into the labeling process. Sources have unknown, task-dependent accuracies and often make systematic labeling errors. Failing to account for these accuracies can negatively impact classification performance. To correct for such label noise, we use data programming[34] to estimate accuracies of each source and ensemble the sources via a label model that assigns a consensus probabilistic label per word.

To learn the label model, $m$-label sources are parameterized as labeling functions $\lambda_1, \ldots \lambda_m$. The vector of $m$-labeling functions applied to $n$ instances forms the label matrix $\Lambda \in \{-1, 0, \ldots, k\}^{m \times n}$. A key finding of data programming is that we can use $\Lambda$ to recover the latent class-conditional accuracy of each label source without ground-truth labels by observing the rates of agreement and disagreement across all pairs of labeling functions $\lambda_i, \lambda_j$[34]. This leverages the fact that while the accuracy $a_i = \mathbb{E}[\lambda_i Y]$ (the expectation of the labeling function output $\lambda_i$ multiplied by the true label) is not directly observable, the product of $a_i a_j = \mathbb{E}[\lambda_i Y \lambda_j Y] = \mathbb{E}[\lambda_i Y]\mathbb{E}[\lambda_j Y]$ is the rate at which labeling functions vote together, which is observable via $\Lambda$. Assuming independent noise among labeling functions, accuracies are then recoverable up to a sign by solving accuracies for disjoint sets of triplets. We refer readers to Ratner et al.[17] for more details.

We use the weak supervision framework Snorkel[11] to train a probabilistic label model that captures the relationship between the true label and label sources $P(\mathbf{Y}, \Lambda)$. Here, the training input is the label matrix $\Lambda$, generated by applying labeling functions $\lambda_1, \ldots \lambda_m$ to the unlabeled dataset D. Formally, $P(\mathbf{Y}, \Lambda)$ can be encoded as a factor graph-based model with $m$-accuracy factors between $\lambda_1, \ldots, \lambda_m$ and our

true (unobserved) label $y$ (Fig. 1, step 3).

$$\theta_j^{Acc}(\Lambda_i, y_i) := y_i \Lambda_{ij} \tag{2}$$

$$p_\theta(\mathbf{Y}, \Lambda) \propto \exp\left(\sum_{i=1}^m \sum_{j=1}^n \theta_j^{Acc} \phi_j^{Acc}(\Lambda_i, y_i)\right) \tag{3}$$

Snorkel implements a matrix completion formulation of data programming, which enables faster estimation of model parameters $\theta$ using stochastic gradient descent rather than relying on Gibbs sampling-based approaches[17]. The label model estimates $P(\mathbf{Y}|\Lambda)$ to provide denoised consensus label predictions $\widehat{\mathbf{Y}}$ and generates our probabilistically labeled dataset $\hat{D}$.

Figure 4 shows how data programming provides a principled way to synthesize a label when there is disagreement across label sources about what constitutes an entity span. The disease mentioning diabetes type 2 is not found in Metathesaurus Names (MTH) or SNOMED Clinical Terms (SNOMEDCT), which leads to disagreement and label errors. Using a majority vote of labeling functions misses the complete entity span, while the label model learns to account for systematic errors made by each ontology to generate a more accurate consensus label prediction.

**Labeling function templates.** In this work, a labeling function $\lambda_j$ accepts an unlabeled sequence $\mathbf{X}_i$ as input and emits a vector of predicted labels $\widetilde{\mathbf{Y}}_{i,j} = (\tilde{y}_{j,1}, \ldots, \tilde{y}_{j,t})$, i.e., a label $\tilde{y}_j \in \{-1, 0, \ldots, k\}$ for each word in $\mathbf{X}_i$. A typical labeling function serves as a wrapper for an underlying, potentially task-specific labeling heuristic such as pattern matching with a regular expression or a more complex rule system. Since these labeling functions are not easily automated and require hand coding, we refer to them as task-specific labeling functions. These are analogous to the rule-based approaches used in 48% of recent medical concept recognition publications[41].

In contrast, medical ontologies can be automatically transformed into labeling functions with little-to-no custom coding by defining reusable labeling function templates. Templates only require specifying a set of target entity categories and providing a collection of terminologies mapped to those categories. These categories are easily derived from knowledge bases such as the UMLS Metathesaurus (where the UMLS Semantic Network[55] provides consistent categorization of UMLS concepts) or other domain-specific taxonomies. In this work, we use UMLS Semantic Groups[56] (mappings of semantic types into simpler, nonhierarchical categories such as disorders) as the basis for our concept categories.

We explore two types of ontology-based labeling functions, which leverage knowledge codified in medical ontologies for term semantic types and synonymy.

Semantic type labeling functions require a set of terms (single or multiword entities) $t \in T$ mapped to semantic types, where a term may be mapped to multiple-entity classes. This mapping is converted to a $k$-dimensional probability vector where $k$ is the number of entity classes $\mathbf{t}_i \to [p_1, \ldots, p_k]$. Given input sequence $\mathbf{X}_i$, use string matching to find all longest-term matches (in token length) and assign each match to its most probable entity class $\tilde{y} = max(\mathbf{t}_i)$, abstaining on ties. Using the longest match is a heuristic that helps disambiguate nested terms (lung as anatomy vs. lung cancer as a disease). Matching optionally includes a set of slot-filled patterns to capture simple compositional mentions (e.g., {*} ({*}) → Tylenol (Acetaminophen)).

Synonym (synset) labeling functions require synsets (collections of synonymous terms) $\{\hat{t}_1, ..., \hat{t}_n\} \in \hat{T}$ and term T mapped to semantic types. Given input sequence $\mathbf{X}_i$ and it's parent context (e.g., document) search for > 1 unique synonym matches from a target synset and label all matches $\hat{y} = max(\mathbf{t}_i)$. This is useful for disambiguating abbreviations (e.g., Duchenne muscular dystrophy → DMD), where a long-form of an abbreviated term appears elsewhere in a document. Matches can be unconstrained, e.g., any tuple found anywhere in a context, or subject to matching rules, e.g., using Schwartz–Hearst abbreviation disambiguation[57] to identify out-of-dictionary abbreviations.

**Training the BioBERT end model**. The output of the label model is a set of probabilistically labeled words, which we transform back into sequences $\hat{D} = \{\mathbf{X}_i, \hat{\mathbf{Y}}_\mathbf{i}\}_{i=1}^N$. While probabilistic labels may be used directly for classification, this suffers from a key limitation: the label model cannot generalize beyond the direct output of labeling functions. Rules alone can miss common error cases such as out-of-dictionary synonyms or misspellings. Therefore, to improve coverage, we train a discriminative end model, in this case a deep neural network, to transform the output of labeling functions into learned feature representations. Doing so leverages the inductive bias of pretrained language models[58] and provides additional opportunities for injecting domain knowledge via data augmentation[59] and multitask learning[60] to improve classification performance.

We use the transformer-based BioBERT[24], a language model fine-tuned on the biomedical text. We also evaluated ClinicalBERT[61] for clinical tasks, and found its performance to be the same as BioBERT. BioBERT is trained as a token-level classifier with a max sequence length of 512 tokens. We follow Devlin et al.[58] for sequence-labeling formulation, using the last BERT layer of each word's head wordpiece token as the contextualized embedding. Since sequence labels may be incomplete (i.e., cases where all labeling functions abstain on a word), we mask all abstained tokens when computing the loss during training. We modified BioBERT to support a noise-aware binary cross-entropy loss function[34] that minimizes the expected value with respect to $\hat{\mathbf{Y}}$ to take advantage of the more informative probabilistic labels.

$$\hat{w} = argmin_w \frac{1}{N} \sum_{i=1}^N \mathbb{E}_{\hat{y} \sim \hat{\mathbf{Y}}}[L(w, x_i, \hat{y})] \tag{4}$$

**Hyperparameter tuning for the label and end models**. All models were trained using weakly labeled versions of the original training splits, i.e., no hand-labeled instances. We used a hand-labeled validation and test set for hyperparameter tuning and model evaluation, respectively. Result metrics are reported using the test set. The label model was tuned for learning rate, training epochs, L2 regularization, and a uniform accuracy prior used to initialize labeling function accuracies. BioBERT weights were fine-tuned, and end models were tuned for learning rate and training epochs. We used a linear decay learning rate schedule with a 10% warmup period. See Supplementary Tables 5 and 6 for hyperparameter grids.

**Metrics**. We report precision, recall, and F1 score for all tasks. DocTimeRela is reported using microaveraging. NER metrics are computed using exact span matching[62]. Each NER task is trained separately as a binary classifier using IO (inside, outside) tagging to simplify labeling function design, with predicted tags converted to BIO (beginning, inside, outside) to properly count errors detecting head words. Span task metrics are calculated assuming access to gold test set spans, as per the evaluation protocol of the original challenges. Label model and BioBERT scores are reported as the mean and standard deviation of 10 runs with different random seeds. A two-sided Wilcoxon signed-rank test with an alpha level of 0.05 was used to calculate statistical significance.

**Reporting summary**. Further information on research design is available in the Nature Research Reporting Summary linked to this article.

## Data availability
All primary data that support the findings of this study are available via public benchmark datasets (BC5CDR, https://biocreative.bioinformatics.udel.edu/tasks/biocreative-v/track-3-cdr/) or are otherwise available per data use agreements with the respective data owners (ShARe/CLEF 2014, https://physionet.org/content/shareclefehealth2014task2/1.0/; THYME, https://healthnlp.hms.harvard.edu/center/pages/data-sets.html; i2b2/n2c2 2009, https://portal.dbmi.hms.harvard.edu/projects/n2c2-nlp/). The data that support the findings of the clinical case study are available on request from the corresponding author J.A.F. These data are not publicly available because they contain information that could compromise patient privacy. Trove requires access to the UMLS, which is available by license from the National Library of Medicine, Department of Health and Human Services, https://www.nlm.nih.gov/research/umls/index.html. Open source ontologies used in this study are available at SPECIALIST Lexicon, https://lsg3.nlm.nih.gov/LexSysGroup/Summary/lexicon.html; Disease Ontology, https://bioportal.bioontology.org/ontologies/DOID; Chemical Entities of Biological Interest (ChEBI), ftp://ftp.ebi.ac.uk/pub/databases/chebi/; Comparative

Toxicogenomics Database (CTD), http://ctdbase.org; AutoNER core dictionary, https://github.com/shangjingbo1226/AutoNER/blob/master/data/BC5CDR/dict_core.txt; ADAM abbreviations database, http://arrowsmith.psych.uic.edu/arrowsmith_uic/adam.html; and the Clinical Abbreviation Recognition and Disambiguation (CARD) framework, https://sbmi.uth.edu/ccb/resources/abbreviation.htm.

## Code availability
Trove is written in Python v3.6, spaCy 2.3.4 was used for NLP preprocessing, and Snorkel v0.9.5 was used for training the label model. BioBERT-Base v1.1, Transformers v2.8[63], and PyTorch v1.1.0 were used to train all discriminative models. Trove is open-source software and publicly available at https://github.com/som-shahlab/trove; https://doi.org/10.5281/zenodo.4497214[64].

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

## Acknowledgements
This work was funded under NLM R01-LM011369-05. Thanks to Birju Patel and Keith Morse who did our COVID-19 clinical annotations and to Daisy Ding and Adrien Coulet who helped refine experimental hypotheses during the early stages of this project. Computational resources were provided by Nero, a shared big data computing platform made possible by the Stanford School of Medicine Research Office and Stanford Research Computing Center. Additional thanks to reader feedback from Stephen Pfohl, Erin Craig, Conor Corbin, and Jennifer Wilson.

## Author contributions
J.A.F. conceived the initial study. J.A.F., E.S., S.K., S.F., J.P. and A.C. wrote code and conducted an experimental analysis of machine-learning models. A.C. and J.A.F. managed and adjudicated clinical text annotations. J.A.F., E.S and N.H.S. contributed ideas and experimental designs. N.H.S. supervised the project. All authors contributed to writing.

## Competing interests
The authors declare no competing interests.
