## [Peer Review File · Nature Communications]

Reviewers' Comments:

Reviewer #1:

Remarks to the Author:

The manuscript presents a framework, Trove, for weakly supervised entity classification models using medical ontologies and expert-generated rules. The manuscript is well written but the contribution of the work needs some better justification.

1. The contribution of the proposed framework is not obvious. How is Trove different from Snorkel?
2. It indicates multiple advances comparing to the previous weak supervision or distant supervision approaches. However, what exactly are those advances? The users still need to choose an ontology's class taxonomy to entity classes. It still depends on the coverage of a broad range of medical ontologies. It still needs to leverage pre-trained language models to deal with out of vocabulary words.
3. The experiment was conducted using public benchmark datasets. In this situation, it would be important to incorporate the state-of-the-art performance on those shared tasks. It is not clear how exactly the comparison of Trove with existing weakly supervised methods was conducted. It is not clear how exactly task specific rules were created. Logically, those specific rules are only meaningful if the performance was reported on a blind test set (which is not the case as those datasets are public benchmark data sets).

Reviewer #2:

Remarks to the Author:

The authors introduce Trove, a novel algorithm based on weak supervision for classifying medical entities in text (named entity recognition) in support of clinical analytics. The work is motivated by the high cost and limited availability of manual annotations for supervised learning algorithms. The authors leverage the terms classified as disorders, chemicals and drugs in biomedical ontologies to create a label model based on the association between words and biomedical terms. The model is used to identify and classify biomedical terms in clinical and compared to a simple soft majority vote model. The authors also use predictions from this model to train the deep learning model, BioBERT, in a weakly supervised manner (compared to human annotations). The 3 approaches, soft majority vote, label model and weakly supervised BioBERT are evaluated through a series of experiments using multiple annotated clinical datasets generated for competitive evaluations, as well as a recent COVID dataset. The authors show that the weakly supervised BioBERT model compares favorably to a fully supervised model, especially when the ontology terms are complemented with task-specific rules. The authors conclude that their approach is a robust and economical alternative to manual annotation for identifying medical entities in clinical text.

This paper addresses the issue of identifying medical entities in text in the absence of a large annotated corpus, which is an important limitation for supervised machine learning algorithms. Circumventing reliance on manual annotation while maintaining good performance is key to supporting clinical analytics.

The authors provide convincing evidence for the validity of their approach by comparing their approach to a fully supervised model (showing the performance of the weakly supervised model comes close), and by evaluating it through a rich set of experiments on multiple reference datasets used in a variety of collaborative evaluations (showing their model generalizes to multiple datasets and tasks).

The results are extremely encouraging, with F1 scores around 76-91% depending on the type of entities.

The manuscript is well written for the most part (see below). A rich set of references is provided.

The manuscript is relatively dense, but remains accessible. The authors make their code available and most of the experiments have been performed on open datasets.

This reviewer, however, has some reservations about this manuscript, essentially regarding the clarity and presentation of the underlying methods (section 2.3). This issues can be addressed easily with limited rewriting of the manuscript.

Overall, this is solid work and this manuscript should establish the use of weakly supervised machine learning leveraging ontologies for medical entity classification and named entity recognition.

Clarity and presentation of the underlying methods

- The presentation of the label model is not intuitive and hard to follow. The illustration provided in Figure 2 does little to explain the label model. Providing examples in section 2.3.2 would do wonders.

- Several elements need to be clarified in section 2.3.3.

- * "the UMLS Semantic Network [50] provides a shared taxonomy for over a hundred medical terminologies" is ambiguous and potentially misleading in this context. The UMLS Semantic Network defines a taxonomy of semantic types and the semantic types are used to categorize the concepts in the UMLS Metathesaurus. In contrast, the sentence above could easily suggest that the Semantic Network organizes the Metathesaurus concepts into a unified taxonomy, which is not the case. Moreover, it is quite unclear what use is made of the Semantic Network in this work. It seems to be that the UMLS concepts are grouped according to the semantic types (or semantic groups), but this is not explicitly stated anywhere.

- * The notion of "taxonomy labeling function" is developed right after introducing the Semantic Network (and the taxonomic organization it provides). However, there is no evidence that the taxonomy labeling function actually relies on any taxonomic relations from the Semantic Network or from the medical ontologies themselves. The examples provided in this paragraph are all unrelated to taxonomy. Overall, it is fairly unclear what the role of a taxonomy is in the so-called taxonomy labeling function.

- * Soft majority vote is never defined. It is only illustrated through Figure 2.

Other comments and suggestions

- The authors do not mention whether they have eliminated from the UMLS Metathesaurus those ontology terms explicitly flagged as "suppressible", because they lack face validity (e.g., "prostate" as a synonym for "prostate cancer" in ICD).

- The UMLS plays a central role in this approach and should be listed as part of the datasets (under Data availability).

- The differences in performance metrics (F1) should be analyzed statistically. It would be interesting to see which differences are statistically significant.

- The acronym SVM should be avoided to eliminate possible confusion with the most common Support Vector Machine.

- It is not entirely clear what the rationale is for the experiment in 3.3.2 or what can be learned from it. Partitioning the UMLS Metathesaurus into sets of ontologies seems beneficial, but it would make sense to base the groupings on creating/avoiding redundancy of terms across sets rather than randomly.

- The discussion would benefit from being organized into subsections.

Grammar and typos

- "orders-of-magnitude more" should not be hyphenated

Reviewer #1 (Remarks to the Author):

The manuscript presents a framework, Trove, for weakly supervised entity classification models using medical ontologies and expert-generated rules. The manuscript is well written but the contribution of the work needs some better justification.

1. The contribution of the proposed framework is not obvious. How is Trove different from Snorkel?

Thank you for the opportunity to clarify our contributions.

Snorkel is a generic framework for modeling the accuracy and structure of labeling sources using a probabilistic label model and is agnostic to the target application. Snorkel does not provide any domain-specific methods or software tooling for training medical entity classifiers or automating the process of designing labeling functions. This is a significant barrier in practice, as creating labeling functions is a non-trivial undertaking without domain expertise.

Trove is an enhancement of Snorkel for the medical text domain, providing a mechanism and toolset for automatically creating labeling functions based on medical ontologies. These labeling functions are easily generated without requiring task-specific coding and are compatible with popular biomedical annotators such as NCBO BioPortal and MetaMap. Quantifying the relative performance benefits of automatically combining multiple medical ontologies is another core contribution of our work. With Trove, we demonstrate the ability to automatically combine biomedical ontologies as supervision sources for improving end model performance.

By publicly releasing Trove to the research community, we hope to speed up the process of comparing, refining, and creating new weakly supervised entity classification models, especially those based on the UMLS. We now describe this in the manuscript's introduction, lines 53-61 . The content added is:

“We present a Trove, a framework for training weakly supervised medical entity classifiers using off-the-shelf ontologies as a source of reusable, easily automated labeling heuristics. Doing so transforms the work of using weak supervision from that of coding task-specific labeling rules to defining a target entity type and selecting ontologies with sufficient coverage for a target dataset -- which is a common interface for popular biomedical annotation tools such as NCBO BioPortal and MetaMap [10,16]. We examine whether ontology-based weak supervision, coupled with recent pre-trained language models such as BioBERT, reduces the engineering cost of creating entity classifiers while matching performance of prior, more expensive, weakly supervised methods. We further investigate how ontology-based labeling functions can be extended when we need to incorporate additional, task-specific rules.”

2. It indicates multiple advances comparing to the previous weak supervision or distant supervision approaches. However, what exactly are those advances?

We have clarified these advances in the manuscript's method section, lines 289-295 The content added is:

“Trove advances weakly supervised medical entity classification by: (1) eliminating the requirement for identifying probable entity spans a priori by combining word-level weak supervision with contextualized word embeddings; (2) developing general purpose, more easily automated ontology-based labeling functions which reduce the need for engineering hand-coded rules; (3) quantifying the relative contributions of sources of label assignment – such as pre-existing ontologies from the UMLS (low cost) and task-specific rule engineering (high cost) – to the achieved performance for a task; and (4) evaluating Trove in a deployed medical setting, tagging symptoms and risk factors of COVID-19.”

The users still need to choose an ontology's class taxonomy to entity classes. It still depends on the coverage of a broad range of medical ontologies. It still needs to leverage pre-trained language models to deal with out of vocabulary words.

In practice, we observe that users routinely need to tailor their entity definitions for specific use cases and we believe that enabling users to select and refine their entity taxonomy is a considerable strength, differentiating Trove from other medical NER approaches.

The reliance on pre-trained language models is also a strength because pre-trained language models are increasingly “plug-and-play” components of NLP pipelines, providing off-the-shelf, commodity models for improving classification accuracy and reducing sample complexity. As pretrained model APIs (e.g., Hugging Face’s Transformers library ¹) continue improving, integrating pre-trained NLP models into workflows will become the norm.

3. The experiment was conducted using public benchmark datasets. In this situation, it would be important to incorporate the state-of-the-art performance on those shared tasks.

We thank the reviewer for their comments and now discuss published state-of-the-art performance in our results section. We note that our fully supervised (FS) baselines were configured to match existing state-of-the-art BioBERT baselines using hyperparameter configurations as reported by Lee et al. ² for BC5CDR. Our fully supervised BioBERT Drug and DocTimeRel supervised results are themselves state-of-the-art performance to the best of our knowledge. We have clarified these advances in the results section and Table 1 of the manuscript, lines 85-88. The content added is:

“For reference, we also included published F1 metrics for state-of-the-art (SOTA) supervised performance for each task, as determined to the best of our knowledge. Note some published SOTA benchmarks (e.g., BC5CDR in Lee et al. [15]) use both the hand-labeled train and validation sets for training, so they are not directly comparable to our experimental setup.”

It is not clear how exactly the comparison of Trove with existing weakly supervised methods was conducted.

Thank you for this suggestion. We have added these experiments and have updated the manuscript (lines 154-157).

“All performance numbers are for models trained on the original training set split, with the exception of SwellShark which is trained on an additional 25,000 weakly labeled documents. All weakly supervised methods use the labeling functions, preprocessing, and dictionary curation methods as described in the original manuscripts.”

It is not clear how exactly task specific rules were created. Logically, those specific rules are only meaningful if the performance was reported on a blind test set (which is not the case as those datasets are public benchmark data sets).

Thank you for this feedback. We have updated the results section to better describe this workflow and provided a case study illustrating the rule creation process in our supplemental materials, Supplementary Fig. 1 and Supplementary Methods.

Only training set data was examined during labeling function development; the test set was held out during all labeling function coding and model tuning (i.e. it was a blinded test set). We acknowledge that a non-public test set is a better way to evaluate model accuracy. Because no public, blinded benchmark dataset exists for clinical named entity classification, we used existing open datasets from which we created blinded test sets from them. Doing so enables researchers to verify and build upon our results.

In addition, we prospectively validate performance of our disorder tagging rules on an independent clinician labeled dataset of Stanford Health Care emergency department notes. No changes were made to the task specific rules, which were developed through inspection of the original ShARe/CLEF MIMIC-II training documents. This “held out dataset” provides some estimate of rule performance when applied to unseen, out-of-distribution training data and evaluated on another out-of-distribution test set.

We have clarified these details in the manuscript, lines 114-140. The content added is:

“We evaluated our ability to supplement ontology-based supervision with hand-coded labeling functions and estimated the relative performance contribution of adding these task-specific rules. All training set documents were preprocessed to tag entities using the ontology-based labeling functions outlined above and indexed to support search queries for efficient data exploration. The design of task-specific labeling functions is a mix of data exploration, i.e., looking at entities identified by ontology labeling functions to identify errors, and similarity search to identify common, out-of-ontology concept

patterns. Only the training set was examined during this process and the test set was held out during all labeling function development and model tuning.

For NER, we used two rule types to label concepts: (1) pattern matching via regular expressions and small dictionaries of related terms (e.g., illegal drugs); and (2) bigram word co-occurrence graphs from ontologies to support fuzzy span matching. Pattern matching comprised the majority of our task-specific labeling functions. While task-specific labeling functions codify generalized patterns not captured by ontologies, we also note that a number of our task-specific labeling functions were necessary due to the idiosyncratic nature of "ground truth" labels in benchmark tasks. For example, in the i2b2/n2c2 drug tagging task, annotation guidelines included more complex, conditional entity definitions, such as not labeling negated or historical drug mentions. We incorporated these guidelines using the Negation and DocTimeRel labeling functions described below. See supplemental materials for a more detailed example of designing task-specific labeling functions.

For span tasks, which classify Negation and DocTimeRel for pre-identified entities, we do not use ontology-based labeling functions directly for supervision. Instead, ontology-tagged entities were used to guide development of labeling functions that search left and right context windows around a target entity for cue phrases. Designing search patterns for left and right context windows is the same strategy used by NegEx/ConText [17,18] to assign negation and temporal status. For Negation, we built on NegEx by adding additional patterns found via exploration of the training documents.

For DocTimeRel we used a heuristic based on the nearest explicit datetime mention (in token distance) to an event mention [19]. Additional contextual pattern matching rules were added to detect other cues of event temporality, e.g., using section headers such as "past medical history" to identify events occurring before the note creation time."

Reviewer #2 (Remarks to the Author):

The authors introduce Trove, a novel algorithm based on weak supervision for classifying medical entities in text (named entity recognition) in support of clinical analytics. The work is motivated by the high cost and limited availability of manual annotations for supervised learning algorithms. The authors leverage the terms classified as disorders, chemicals and drugs in biomedical ontologies to create a label model based on the association between words and biomedical terms. The model is used to identify and classify biomedical terms in clinical and compared to a simple soft majority vote model. The authors also use predictions from this model to train the deep learning model, BioBERT, in a weakly supervised manner (compared to human annotations). The 3 approaches, soft majority vote, label model and weakly supervised BioBERT are evaluated through a series of experiments using multiple annotated clinical datasets generated for competitive evaluations, as well as a recent COVID dataset. The authors show that the weakly supervised BioBERT model compares favorably to a fully supervised model, especially when the ontology terms are complemented with task-specific

rules. The authors conclude that their approach is a robust and economical alternative to manual annotation for identifying medical entities in clinical text.

This paper addresses the issue of identifying medical entities in text in the absence of a large annotated corpus, which is an important limitation for supervised machine learning algorithms. Circumventing reliance on manual annotation while maintaining good performance is key to supporting clinical analytics.

The authors provide convincing evidence for the validity of their approach by comparing their approach to a fully supervised model (showing the performance of the weakly supervised model comes close), and by evaluating it through a rich set of experiments on multiple reference datasets used in a variety of collaborative evaluations (showing their model generalizes to multiple datasets and tasks).

The results are extremely encouraging, with F1 scores around 76-91% depending on the type of entities. The manuscript is well written for the most part (see below). A rich set of references is provided. The manuscript is relatively dense, but remains accessible. The authors make their code available and most of the experiments have been performed on open datasets. This reviewer, however, has some reservations about this manuscript, essentially regarding the clarity and presentation of the underlying methods (section 2.3). This issues can be addressed easily with limited rewriting of the manuscript.

Overall, this is solid work and this manuscript should establish the use of weakly supervised machine learning leveraging ontologies for medical entity classification and named entity recognition.

We thank the reviewer for their positive comments on our work and support of open software and reproducible research using open data.

Clarity and presentation of the underlying methods

- The presentation of the label model is not intuitive and hard to follow. The illustration provided in Figure 2 does little to explain the label model. Providing examples in section 2.3.2 would do wonders.

Thank you for this feedback. We have updated the methods section (lines 326-359) to better describe the label model and now include a running example to help build intuition for the reader.

- Several elements need to be clarified in section 2.3.3.

* "the UMLS Semantic Network [50] provides a shared taxonomy for over a hundred medical terminologies" is ambiguous and potentially misleading in this context. The UMLS Semantic Network defines a taxonomy of semantic types and the semantic types are used to categorize the concepts in the UMLS Metathesaurus. In contrast, the sentence above could easily suggest

that the Semantic Network organizes the Metathesaurus concepts into a unified taxonomy, which is not the case.

Thank you for pointing out this ambiguity. We have rewritten this sentence based on your comments to accurately reflect how the UMLS Semantic Network can be used to categorize concepts in the UMLS Metathesaurus (lines 369-371).

“These categories are easily derived from knowledge bases such as the UMLS Metathesaurus (where the UMLS Semantic Network [55] provides a consistent categorization of UMLS concepts) or other domain-specific taxonomies.”

Moreover, it is quite unclear what use is made of the Semantic Network in this work. It seems to be that the UMLS concepts are grouped according to the semantic types (or semantic groups), but this is not explicitly stated anywhere.

We have updated the manuscript to better outline how we use the Semantic Network to define types of entities (lines 371-373).

“In this work, we use UMLS Semantic Groups [56] (mappings of semantic types into simpler, non-hierarchical categories such as disorders) as the basis for our concept categories.”

* The notion of "taxonomy labeling function" is developed right after introducing the Semantic Network (and the taxonomic organization it provides). However, there is no evidence that the taxonomy labeling function actually relies on any taxonomic relations from the Semantic Network or from the medical ontologies themselves. The examples provided in this paragraph are all unrelated to taxonomy. Overall, it is fairly unclear what the role of a taxonomy is in the so-called taxonomy labeling function.

We thank the reviewer for this feedback on our imprecise use of the term “taxonomy”, which we used to loosely refer to a categorization of concepts. The reviewer is correct that we do not explicitly utilize taxonomic relations in our labeling functions. We have renamed “taxonomy labeling function” to “semantic type labeling function” to make its design more clear to readers. Our goal in this work was to develop a conservative baseline estimate of performance using only semantic type information.

We have made these changes in the methods section of the manuscript (lines 374-375). The content modified is:

“We explore two types of ontology-based labeling functions, which leverage knowledge codified in medical ontologies for term semantic types and synonymy.

Semantic type labeling functions require a set of terms...”

* Soft majority vote is never defined. It is only illustrated through Figure 2.

Thank you for catching this oversight. We have updated the methods section (lines 330-331, Equation 1) to clearly define majority vote. We now clearly outline each of the three methods used in our main experiments in the results section (lines 78-84).

Other comments and suggestions

- The authors do not mention whether they have eliminated from the UMLS Metathesaurus those ontology terms explicitly flagged as "suppressible", because they lack face validity (e.g., "prostate" as a synonym for "prostate cancer" in ICD).

We did not incorporate information on term types (TTY) or suppressibility of term types given a source terminology (SAB). Our approach was to impose as few assumptions as possible when importing UMLS terminologies, since non-UMLS ontologies typically do not include suppressible term information and the UMLS itself likely contains many terms that lack face validity but are not explicitly coded as suppressible.

We have updated the methods section to describe the UMLS term preprocessing and filtering choices (lines 310-315).

"We applied minimal preprocessing to all source ontologies, filtering out English stopwords [48], applying a letter case normalization heuristic to preserve abbreviations, and removing all single character terms. We did not incorporate UMLS term type information, such as filtering out terms explicitly denoted as "suppressible" within a terminology, since this information is not typically available in non-UMLS ontologies. Our overall goal was to impose as few assumptions as possible when importing terminologies, evaluating their ability to function as "plug-and-play" sources for weak supervision."

- The UMLS plays a central role in this approach and should be listed as part of the datasets (under Data availability).

We have now updated the data availability section to properly reference the UMLS.

- The differences in performance metrics (F1) should be analyzed statistically. It would be interesting to see which differences are statistically significant.

We thank the reviewer for their comments. We have increased the number of model replicates to 10 and now report the statistical significance of pairwise differences in model performance using a Wilcoxon signed rank test.

- The acronym SVM should be avoided to eliminate possible confusion with the most common Support Vector Machine.

We apologize for this confusion. SVM was a transposition typo for soft majority vote (SMV). We have now simplified the manuscript's notion to refer to this baseline method as "majority vote" (MV) to eliminate possible confusion.

- It is not entirely clear what the rationale is for the experiment in 3.3.2 or what can be learned from it. Partitioning the UMLS Metathesaurus into sets of ontologies seems beneficial, but it would make sense to base the groupings on creating/avoiding redundancy of terms across sets rather than randomly.

We thank the reviewer for their comments. The primary purpose of this experiment was to estimate the baseline performance of UMLS ontologies as "plug-and-play" weak supervision sources, with minimal post-processing and user input specifying which terminologies to include as label sources.

Therefore, we picked a simple ranking rule for deterministically including UMLS terminologies (i.e., selecting the top s terminologies sorted by the number of term matches on the unlabeled corpus). The reviewer is correct that users may manually define different groupings of UMLS ontologies or otherwise filter terms based on application-specific domain knowledge and that these groupings may result in better performance (lines 163-166).

"Biomedical annotators such as NCBO BioPortal require selecting a set of target ontologies/terminologies to use for labeling. Since Trove is capable of automatically combining noisy terminologies, given a shared semantic type definition, we tested the ability to avoid selecting specific UMLS terminologies for use as supervision sources."

- The discussion would benefit from being organized into subsections.

We have reorganized the discussion section to have more structure.

Grammar and typos

- "orders-of-magnitude more" should not be hyphenated

Thank you for catching this typo.

References

1. Wolf, T. *et al.* HuggingFace's Transformers: State-of-the-art Natural Language Processing. *arXiv e-prints* arXiv:1910.03771 (2019).
2. Lee, J. *et al.* BioBERT: a pre-trained biomedical language representation model for biomedical text mining. *Bioinformatics* (2019) doi:10.1093/bioinformatics/btz682.

Reviewers' Comments:

Reviewer #1:

Remarks to the Author:

Thanks for the detailed responses including the clarification of the contribution. The Trove framework combines the ontology lookup within a weak supervision framework which can significantly speed up the development process of NER tasks. This reviewer has no further comments but does recommend the citation of one latest work published in JBI which confirms the findings reported in the manuscript:

<https://pubmed.ncbi.nlm.nih.gov/32814201/>

Reviewer #2:

Remarks to the Author:

The issues pointed out by this reviewer have been addressed to satisfaction. Good paper. No further comments.

Reviewer #1 (Remarks to the Author):

Thanks for the detailed responses including the clarification of the contribution. The Trove framework combines the ontology lookup within a weak supervision framework which can significantly speed up the development process of NER tasks. This reviewer has no further comments but does recommend the citation of one latest work published in JBI which confirms the findings reported in the manuscript:

<https://pubmed.ncbi.nlm.nih.gov/32814201/>

Thank you for identifying this related work. We now cite Peterson et al. 2020 in the manuscript's related work section.

“Recent work has explored learning the accuracies of sources to correct for label noise when using rule-based systems to generate training data for text classification [4,35] Weakly supervised clinical applications have explored document and relation classification using task-specific rules [36,37] or leveraging dependency parsing and compositional grammars to automate relation classification for standardizing clinical concepts [38].”

Reviewer #2 (Remarks to the Author):

The issues pointed out by this reviewer have been addressed to satisfaction. Good paper. No further comments.

We thank the reviewer for their prior feedback.